# Study on Cracking Law of Earthen Soil under Dry Shrinkage Condition

**DOI:** 10.3390/ma15238281

**Published:** 2022-11-22

**Authors:** Shaohua Zhang, Jianwei Yue, Xuanjia Huang, Limin Zhao, Zifa Wang

**Affiliations:** 1Zhumadian Highway Administration of Henan Province, Zhumadian 463000, China; 2School of Civil Engineering and Architecture, Henan University, Kaifeng 475004, China; 3Kaifeng Key Laboratory for Restoration and Safety Evaluation of Immovable Cultural Relics, Kaifeng 475004, China

**Keywords:** earthen soil, microscopic model, dry shrinkage, SEM image

## Abstract

Earthen sites are easily eroded by the natural environment, resulting in a large number of micro cracks on the surface. In order to explore the internal relationship between environmental factors and the cracking law of soil sites, this paper carries out dry shrinkage tests of different soil layers at the Zhouqiao site, reconstructs the study on cracking law of earthen soil under dry shrinkage-conditioned microstructure of site soil at different depths based on electron microscope pictures and finite element method, and explores the influence of different moisture content on the cracking of soil samples at the site. The results show that under conditions of dry shrinkage, the thickness of the soil layer has the greatest influence on the cracking of site soil samples. Due to the internal water loss and shrinkage of the soil sample, the thinner the soil layer, the more often the soil layer cracks first. The crack rate of the soil sample with a thickness of 1 cm is nearly three times higher than that of the soil sample with a thickness of 5 cm. Through numerical simulation analysis, it is found that the evolution process of soil fractures at the Zhouqiao site is mainly divided into the formation stages of initial stress field, single main fracture, secondary fracture and fracture network. The formation time of the secondary fracture is longer than that of the initial stress field and single main fracture, and the cracking of the upper soil sample is more serious than that of the lower soil sample. Under conditions of dry shrinkage, the particle arrangement of the soil sample is relatively loose, and there are many cracks inside, which provides evaporation and infiltration channels for water, forming unrecoverable weak pores, and finally, the cracks start to sprout at the weak points. The research results provide some reference for the disease mechanism and safety analysis of earthen sites.

## 1. Introduction

Earthen ruins are ancient structures made up of earthen materials and have important historical, cultural and scientific values. In recent years, with the rapid development of urban construction, buried earthen sites have been continuously excavated for protection and display through the establishment of museums. Under the protection of museum buildings, although the direct threat of wind erosion and rain on the earthen sites is avoided, the ground elevation of the earthen sites in the site museum is usually lower than the surrounding surface, and under the capillary action, the site proper is directly subjected to the reciprocal action of groundwater, leading to various diseases such as surface cracking, chalking and crusting (Figure 1) [1,2]. The cracking pattern and deterioration mechanism of the site body caused by the disaster-causing effect of shrinkage cracking has become an urgent engineering problem to be solved in preventive conservation work [3].

In recent years, domestic and foreign scholars have obtained research results on the mechanical properties and deterioration characteristics of geotechnical materials [4,5]. Soil sites have been found to be susceptible to erosion by the natural environment, and a large number of cracks are commonly developed on the surface, of which dry shrinkage is a common influencing factor [6]. However, the mechanical mechanism of cracking caused by dry water loss in soils has not been consistently investigated [7]. It is generally believed that the suction force generated by the drying process causes the soil to shrink, and when the shrinkage deformation is restricted, tensile stress will be generated inside the soil, and when the tensile stress exceeds the tensile strength of the soil, cracks will be created [8]. Zhang et al. [9] analyzed three crack characterization parameters, crack width, depth and surface crack area rate, using iodine-starch staining tracer tests combined with digital image processing techniques. Jia et al. [10] used molecular dynamics to simulate the deformation and cracking process of nanoclay particles under water impact, and found that the shedding of SiO_2_ tetrahedral units led to cracks or holes, which also provided a molecular explanation for the microscopic mechanism and process of rupture of geotechnical materials. Wu et al. [11] conducted evaporation tests on clay soils while monitoring shrinkage cracking and hygrothermal evolution, and found that the large difference in suction between the topsoil and the deep soil is an important reason for the uneven shrinkage of the soil at different depths. Xu et al. [12] applied fracture mechanics methods to analyze the formation and evolution of fractures, and to determine if and when soils rupture. Tang et al. [13] quantified microcrack parameters based on the digital image method and explained cracking mechanisms, such as sprouting, extension, bifurcation and merging of cracks. Most of the past studies have focused on the analysis of cracking evolution patterns of soils [14,15,16], and few scholars have conducted dry shrinkage tests in conjunction with changes in moisture content, suction, temperature and cracking [17].

In order to explore the intrinsic connection between environmental factors and cracking patterns of earth sites, and to provide a basis for the research and design of conservation measures for earth sites in dry and wet environments, we also need to know the change patterns of soil samples at the microscopic scale, and to establish the connection between macroscopic and microscopic scales. Therefore, this paper investigates the cracking of the site soil samples at different depths by conducting dry shrinkage tests on the Zhouqiao site with different soil thicknesses, and reconstructing the site soil microstructure at different depths based on electron microscope images and finite element method. The results of the study provide some insight into the disease mechanism and safety analysis of soil sites.

## 2. Research Methodology

### 2.1. Basic Physical Properties of Site Soils

By sampling three different parts of the upper, middle and lower layers of the Zhouqiao site (Figure 1), in accordance with the Standard for Geotechnical Test Methods (GB/T50123-2019), the basic physical properties of the soil in different parts were measured (Table 1). From the data in Table 1 and Figure 1, it can be seen that there are some differences in the water content and cracking of the site soil with the change of depth. Therefore, the depth of the soil layer is an important factor considered in this paper.

### 2.2. Dry Shrinkage Test

The State Bridge site is an open preservation, and the ground elevation of the earthen site is usually lower than the surrounding ground surface. Under the capillary water action, the site body is directly subject to the reciprocal action of groundwater, and the migration of water has caused some impact on the stability of the site body. Due to the long daylight hours and high average temperatures in summer, the site is subject to dry shrinkage, and the surface layer is seriously weathered, chalked and spalled (Figure 1). Therefore, it is crucial to investigate the variation of site soil cracking in depth and under dry shrinkage conditions as outlined by the Site Ontology Safety Evaluation.

In this paper, soil thickness (A) and water content (B) are taken as the factors to be considered in the test group. Since there are water pits in the middle of the site and the moisture content of the soil has reached the liquid limit, the variable values of the test moisture content were taken as 35% of the liquid limit moisture content of the soil sample, 20% of the plastic limit moisture content and the average value of both. To explore the development of the depth of soil damage at the site under dry shrinkage conditions, the thickness of the soil layers of 1 cm, 3 cm and 5 cm were considered in this test. The specimen numbers and test protocols are shown in Table 2; three parallel specimens were made for each group, giving a total of 27.

The test procedure is as follows: ① Dry the soil samples to be used in the oven until the weight. ② Then, the soil samples are configured in the molds of L × W × W = 27 cm × 17 cm × 10 cm, respectively, according to Table 2. ③ Put the soil sample maintained for 24 h into the model KD-2P-80 constant temperature and humidity test chamber for dry shrinkage test (temperature 27 °C, humidity 70%). ④ Stop the test when the moisture content of the specimen reaches the average natural moisture content. ⑤ Take images of specimen surface cracking and calculate fractality using Image-Pro Plus (IPP) 6.0 software. ⑥ Conduct sampling for electron microscopy (SEM) images.

First, regarding the value of water content, this paper takes the protection of the newly excavated Zhouqiao site as the research object. The groundwater level in Kaifeng is high, and the water content of the newly excavated Zhouqiao site soil is far higher than the current site (natural water content). Serious cracks have occurred on the surface of the site since excavation. Orthogonal tests under the existing water content do not conform to the change of the site from the high water content during excavation to the low water content after cracking. The process of site excavation is considered in the test design. Second, the Zhouqiao site studied in this paper is an open preservation site, and the ground elevation of the earthen site is usually lower than the surrounding surface. Under the action of capillary water, the site body is directly subject to the reciprocating action of groundwater, resulting in the surface cracking of the site. Third, being affected by rainfall, the water content of the site varies at different depths. Specifically, there is a water pit in the middle of the site, and the water content of the soil has reached the liquid limit, which is also considered in the test design [18]. Therefore, in order to consider the change of the actual water content of the site and the most adverse environmental effects, the plastic limit, liquid limit and their average values are selected in the orthogonal test.

Regarding the value of soil layer thickness, the Zhouqiao site is an open preservation site, and the ground elevation of the earthen site is usually lower than the surrounding surface. The site is subject to drying shrinkage, and the surface is seriously weathered and pulverized. In the early stages of field investigation, the research team used ammunition. A plastic rod (insert an elastic plastic rod with a diameter of 1 mm into each crack until obvious resistance is felt) was used to obtain statistics on the surface crack depth of the earthen site. It was found that the surface crack depth of the Zhouqiao site was mostly within the range of 1 cm to 5 cm. Therefore, soil layer thicknesses of 1 cm, 3 cm and 5 cm are considered.

### 2.3. Numerical Modeling

By applying IPP software to SEM photos at 500× magnification, the authors propose a numerical modeling method based on an image reconstruction method that can truly reflect the microstructure of soil samples [19]. In this paper, the SEM maps (Figure 2) of the surface soil, middle soil and deep soil are modeled in layers (Figure 3), using this method to carry out cracking simulation analysis of soil samples.

The model element type is CPE4R, the mesh number of the lower soil sample model is 1836 and the mesh number of the upper platen is 234. Regarding the convergence process, and because the material will have a nonlinear property during the numerical simulation, when the load gradually approaches the ultimate bearing capacity, if the given load step is too large, it will lead to iterative divergence and convergence difficulties. To achieve this, an explicit dynamics analysis is used [20].

The constraint conditions of the model are: the lower boundary of the model is constrained in the x direction, y direction and rotation angle.

A two-dimensional model with a length × width of 0.644 mm × 0.431 mm was established using ABAQUS software 2016, and the numerical model consisted of three phases of soil particles, pores and cement, and the material property parameters were defined according to previous research results obtained by [19]. The simulation model is based on the mechanism of surface tension, and the area force is chosen to apply the load (Figure 4). In considering the open soil, the average particle size of the studied powder soil particles is 0.074 mm in this paper, and the relationship between the surface tension of water σ (N/m) and temperature t (°C) at atmospheric pressure can be expressed, taking the surface tension coefficient of pure water at temperature 20 °C, T = 7.284 × 10 ^−5^ N/mm, and the contact angle is taken as 10°. By fitting the image to Equation (1) [21], the final graph of inter-pore force versus water content is obtained (Figure 5) [21].
F = Fc + Ts
= πα(Rsinφ)^2^(1/r1 − 1/r2) + 2πRαsinφ
= πRαsinφ{sinφ[cos(φ + θ)/(1 − cosφ) − cos(φ + θ)/[sin(φ + θ) + cosφ − sinθ − 1]] + 2}(1)
where r1, r2 are the two radii of curvature of the reciprocal anti-bending lunar surface; θ is the contact angle between water and solid surface; φ is the angle of maximum coverage of the water film at the curved liquid surface on spherical soil particles, i.e., the saturation angle; α is the surface tension coefficient.

After the finite element model is established, the displacement constraints are applied to the lower left and right boundaries of the model. The model mesh cell type is CPE4R, which is a four-node bilinear plane strain quadrilateral cell with reduced integration. The advantage of using this cell type is that when it comes to mesh distortion problems, especially large mesh deformation, a linear reduced integration cell with a fine mesh profile can be used. Due to the non-linear nature of the material when performing numerical simulations, when the load gradually approaches the ultimate bearing capacity, if the given load step is too large, it will lead to iterative scattering and convergence difficulties. For this purpose, analysis is performed using explicit dynamics. Finally, three numerical models for dry shrinkage tests were established according to Figure 2 (Figure 6).

## 3. Analysis of Test Results

### 3.1. Analysis of Dry Shrinkage Test Results

Figure 7 shows the results of the dry shrinkage test, as shown in Figure 7. There are large differences in the cracking of soil samples under different test schemes, among which the soil samples in group A_1_B_1_ are the most seriously cracked, while the soil samples in group A_3_B_3_ are basically uncracked. Table 3 shows the results of the cracking rate of the dry shrinkage test; it can be obtained that the order of the cracking rate of the dry shrinkage test is A_1_B_1_ > A_1_B_2_ > A_1_B_3_ > A_2_B_3_ > A_2_B_1_ > A_2_B_2_ > A_3_B_1_ > A_3_B_2_ > A_3_B_3_, and the cracking rate of the soil sample of the A_1_B_1_ group is much larger than the other eight groups, and the cracking rate of the soil sample reaches the highest 9.1% in the nine groups of the test; higher than the second placed A_1_B_2_ group 1.2, and 7.9 higher than the lowest group A_3_B_3_, indicating that the effect of different factors on the effect of site soil cracking can be seen from the test. Combining the cracking pictures of the specimens in Figure 7 with the results of the fracture rate in Table 3, it can be concluded that the degree of soil cracking at the site is inversely proportional to the thickness of the soil layer; the thinner the soil layer, the more the soil layer cracks first and presents a staggered fracture network. For example, the fracture rates of the specimens in the three groups A_1_B_1_, A_1_B_2_ and A_1_B_3_ are generally higher than those in the other six scenarios, with a macroscopic expression of a staggered fracture network. As the thickness of the site soil layer increases, only single, coarse main fissures appear in the soil layer; for example, the lowest fissures are found in specimens from the A_3_B_1_, A_3_B_2_ and A_3_B_3_ groups, which are nearly three times lower than the fissures of specimens from the other six scenarios.

The reason for the above test phenomenon is that, under dry shrinkage conditions, the soil sample loses water and internally shrinks, the water between soil particles becomes less and less, the pore size between soil particles decreases and the compactness increases by surface tension, and the soil sample will be the first to sprout fissures in the pores between particles, providing a channel for the migration of water. As water continuously migrates from the interior of the soil sample to the surface layer of the soil sample and evaporates into the atmosphere, the soil sample severely loses water, which eventually leads to cracking of the soil sample and the formation of a fissure network. As can be seen in Figure 7, the initial water contents presented in Figure 7a,d,g are all 20%, and the internal water content of the soil sample is lower, which is more likely to produce fissures under the same conditions.

### 3.2. Microstructure Analysis

Figure 8 shows the SEM results of the dry shrinkage test. From Figure 8, it can be seen that, under the dry shrinkage condition, the association between the soil particles suffers damage and the distance between the particles increases, and even some of the particles in the cluster are separated and dislodged, so that more fissures are produced in the soil. The fissures are widely distributed, the soil particles are loosely arranged, the gap between the particles is large, the size of the soil particles is not uniformly distributed and the shape of the particles is mostly plate-like and spherical; there is only geometric accumulation of soil particles between the clay particles, and there is no solid connection. It can be considered that the surface deterioration of soil sample under the effect of dry shrinkage is the comprehensive manifestation of microstructural changes of particles, pores and cements in the soil sample caused by water loss and shrinkage of hydrophilic clay minerals [22].

The SEM photos with 500× magnification were processed by applying IPP software to obtain the SEM map pore area ratios (Table 4). As seen in Table 4, the order of the size of the fractality ratio of the microscopic electron microscopy map of the dry shrinkage test was A_1_B_1_ > A_1_B_2_ > A_1_B_3_ = A_2_B_3_ > A_2_B_1_ > A_2_B_2_ > A_3_B_1_ > A_3_B_2_ > A_3_B_3_, and it was found that the size law of the fractality area ratio shown by the microstructure of the specimen was basically consistent with the size law of the macroscopic fractality ratio (Table 3). Combining Figure 8 and Table 4, it can be obtained that the smaller the specimen thickness is, the larger the fracture ratio is and the more inter-particle pores there are. Damage cracks provide evaporation and infiltration channels for water, and a large amount of water will be transported to the surface layer of the soil sample and will cause the relative humidity of the internal pores of the soil sample to significantly decrease, causing the internal pores of the soil sample to continuously shrink, causing irregular moisture migration paths and the soil around the pores will first occur by dry shrinkage phenomenon.

### 3.3. Analysis of Numerical Simulation Results

Figure 9 shows the damage clouds under different stages of model 1, and Figure 10 shows the damage clouds of models 1–3 compared with the test results. The blue part in Figure 9 is the cement; the irregular white parts are the soil particles; the round white parts are the pore space; and the red parts are the damage fracture. Due to the limited space and the similarity of the work, this paper mainly analyzes the fracture evolution process with the simulation results of model 1. By analyzing the damage cloud diagram under different stages of model 1, it is found that the fracture evolution process is mainly divided into the formation stage of initial stress field, the formation stage of single primary fracture, the formation stage of secondary fracture and the formation stage of fracture network.

In the formation stage of the initial stress field, soil samples in the drying process as the moisture content decreases, the pore water distributed between the soil particles continuously migrate and exclude, resulting in the soil particles by the action of surface tension, which leads the pore surface to produce irrecoverable shrinkage deformation; the pore around the stress concentration phenomenon easily leads to the occurrence of damage. When the tensile stress is greater than the cohesive force of the soil sample, the initial stress field will be formed on the surface of the inter-particle pore, at which time the fracture begins to sprout, as shown in Figure 9b. In the formation stage of a single main fracture, it can be found by numerical simulation (Figure 9b) that as the stress concentration phenomenon occurs locally in the soil sample, when the magnitude of tensile stress exceeds the tensile strength between soil particles, fractures will sprout at the weak point of intergranular pore space of the soil sample, and as the dry shrinkage test continues, the soil sample starts to produce single main fractures at each weak point and the soil sample “fragile surface” is continuously extended, as shown in Figure 9c. At the stage of secondary fracture formation, it was found by the numerical simulation results (Figure 9c) that the microcracks no longer singularly developed as the damage microcracks further extended and aggregated. After the first crack is created, the stress concentration at the extension end of the crack will cause a new tensile stress field to be generated on the vulnerable crack surface, and when the displacement of soil particles in its path is restricted or stress concentration occurs, new cracks will be derived, and each new crack will cause the release of the accumulated tensile stress on the surface and then expand and gradually penetrate along the edge of soil particles, and finally form several cracks that penetrate the whole specimen. Combined with the macroscopic test results, it is found that crack development is the fastest at this stage, and secondary cracks will stop extending when they intersect with the adjacent main cracks; however, the width of secondary cracks will continue to increase. When the width value of the secondary fracture is close to the main fracture, new fractures will continue to be derived at the weak point of the fracture. In the formation stage of the fracture network, by combining the numerical simulation results (Figure 9d) and the macroscopic test results, it is found that the newly derived fractures will converge to the generated fractures. As the primary and secondary fissure paths keep close together, the soil of the state bridge site will be eventually divided into several areas to form a fissure network.

By applying IPP software to measure the fracture rate of the simulation results (Figure 10), the numerical model fracture rate results of the three parts were 18%, 15% and 12%, and it was found that the fracture rate of the upper soil sample was higher, while the lower soil sample had a lower fracture rate. Combined with 3.2 microstructure analysis, it can be seen that, as the water in the lower soil sample continuously migrated to the surface layer, the internal pores of the soil sample were subjected to surface tension and produced irrecoverable fissures, leading to more obvious cracking in the upper soil sample.

Figure 11 shows the axial displacement clouds under different stages of model 1. By analyzing Figure 11, it is found that the displacement of the soil sample regularly changes, and the overall displacement shows a stratified change in which the bottom displacement is the smallest, while the upper displacement is larger. In the process of drying and shrinking, the displacement distribution inside the specimen is uneven, and the local displacement will abruptly change. Combined with the macroscopic test results, it is found that the secondary fracture is the fastest to develop and, in the numerical simulation analysis, it is found that the local displacement abruptly changes at Step Time = 0.52, and the displacement is in the most part of 1.580 × 10^−4^~3.951 × 10^−4^ mm (mainly in the sky blue area in the figure), when Step Time = 0.37, the displacement change amplitude of 5.5985 × 10^−5^~1.120 × 10^−4^ mm is close to 3 times, indicating that secondary fracture development is faster than the initial stress field formation. Figure 12 shows the axial displacement clouds of models 1–3. Through analysis of the axial displacement clouds of the three models in Figure 12, it is found that the cause of soil sample cracking is due to the inhomogeneity of soil sample particle distribution, and with the continuous migration of pore water, the surface tension of intergranular pores continuously changes, and when the initial stress inside the soil sample exceeds the cohesive force, rubbing occurs between the soil sample particles and the cement, and relative displacement is generated, leading to soil sample sprouting fissures. The sprouted fissure will become a migration channel for the pore water to reach the surface of the soil sample, resulting in irrecoverable deformation of the soil sample pore, which is the reason why the upper soil sample cracks before the lower soil sample.

## 4. Gridding

After the components are built, in order to ensure the correctness of the model, it is necessary to mesh the model first, so as to avoid invalid load, contact and boundary conditions caused by sharp corners and unclosed lines in the model, and to ensure the convergence and correctness of the model in calculation and analysis. The operations conducted in abaqus software are as follows: select mesh at module; enter verify mesh; click highlight at analysis checks, and you can observe whether there are errors in the established model; if there are errors, the model will display pink; in the finite element analysis, the size and density of the mesh determine the accuracy and efficiency of the calculation results; therefore, when meshing, it is necessary to reduce the grid density without affecting the simulation results to save model analysis time. Based on the analysis of the above influencing factors, the grid division process using abaqus software is as follows: ① select the “assign element type” function to assign the cell type function, and select the area to be divided into grids; ② at family, set the family to plane strain, and ③ set the cell library to explicit; ④ click the seed part function, and the approximate global size is set to 0.01; at this time, the grid density has the highest calculation efficiency on the premise of ensuring the calculation accuracy; ⑤ click the assign mesh controls function, select the area to be divided into meshes, select quadrilateral as the cell type, free as the technique, and advancing front as the algorithm; ⑥ finally, click the mesh part to divide the mesh. The grid generation results are shown in Figure 13. [23]

## 5. Conclusions

In this paper, we investigated the cracking of the site soil samples at different depths by conducting dry shrinkage tests with different soil thicknesses at the Zhouqiao site and reconstructing the microstructure of the site soil at different depths based on electron microscope images and finite element method. The main conclusions were as follows:

(1) Under dry shrinkage conditions, the thickness of the soil layer has the greatest effect on the cracking of soil samples at the site. Due to the water loss and shrinkage inside the soil sample, the thinner the soil layer, the more the soil layer cracked first, and the cracking rate of the soil sample with a thickness of 1 cm was nearly three times higher than the cracking rate of the soil sample with a thickness of 5 cm. The cracking degree of the site soil layer is inversely proportional to the thickness of the soil layer. The thinner the soil layer is, the more the soil layer cracks first and presents a staggered fracture network.

(2) Through numerical simulation analysis, it is found that the soil fracture evolution process of the Zhouqiao site is mainly divided into four stages: initial stress field, single main fracture, secondary fracture and formation of fracture network. The formation stage of secondary fissures is longer than the formation stage of the initial stress field and single main fissure, and the upper soil sample cracking is more serious than the lower soil sample cracking.

(3) The smaller the sample thickness is, the greater the crack rate is, and the more pores between particles there are; the soil around the pores will shrink first.

(4) Soil samples under dry shrinkage conditions produce more internal fissures and looser particle arrangements, which provide evaporation and infiltration channels for water, forming irrecoverable weak pores, and eventually, fissures will begin to sprout at the weak points.

## Figures and Tables

**Figure 1 materials-15-08281-f001:**
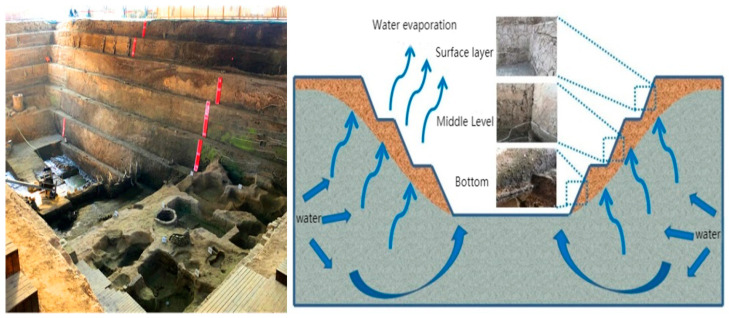
Schematic diagram of working conditions of the Zhouqiao site under dry shrinkage conditions.

**Figure 2 materials-15-08281-f002:**
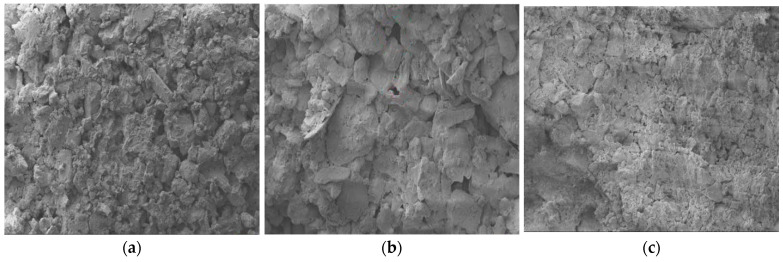
SEM images. (**a**) SEM image of upper soil; (**b**) SEM image of middle soil; (**c**) SEM image of subsoil.

**Figure 3 materials-15-08281-f003:**
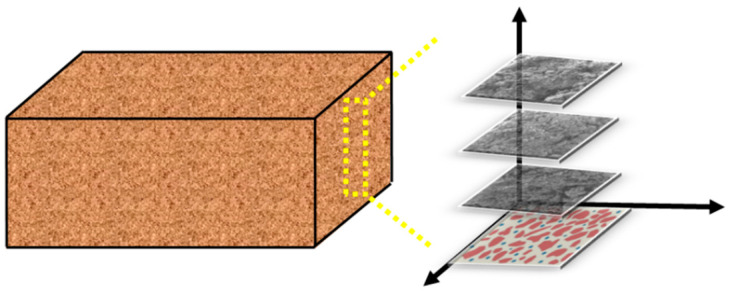
Layered modeling diagram.

**Figure 4 materials-15-08281-f004:**
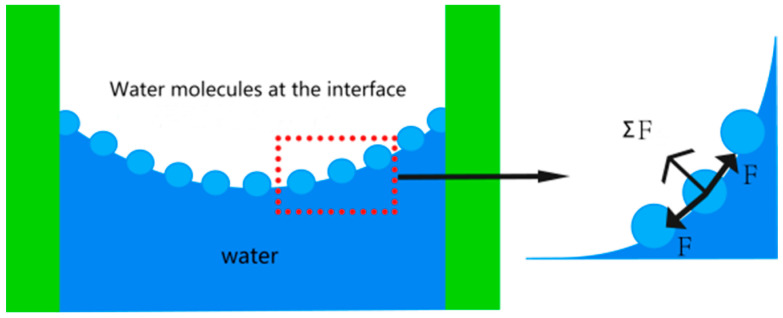
Mechanism of surface tension.

**Figure 5 materials-15-08281-f005:**
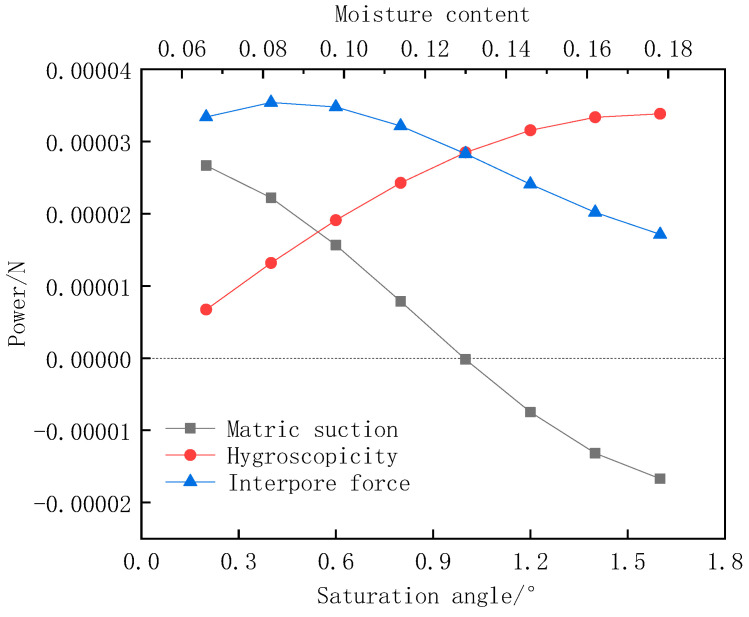
Relationship between pore stress and water content.

**Figure 6 materials-15-08281-f006:**
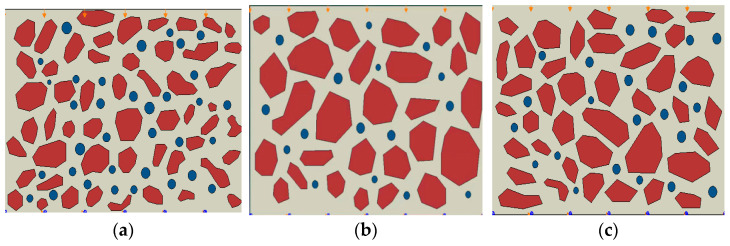
Simulation models. (**a**) Simulation model 1; (**b**) Simulation model 2; (**c**) Simulation model 3.

**Figure 7 materials-15-08281-f007:**
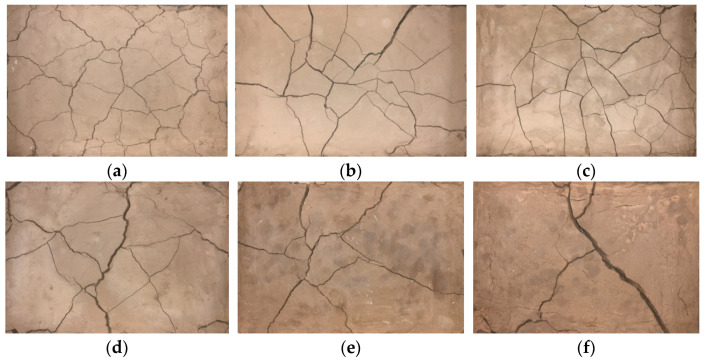
Pictures of dry shrinkage test results. (**a**) A_1_B_1;_ (**b**) A_1_B_2;_ (**c**) A_1_B_3;_ (**d**) A_2_B_1;_ (**e**) A_2_B_2;_ (**f**) A_2_B_3;_ (**g**) A_3_B_1;_ (**h**) A_3_B_2;_ (**i**) A_3_B_3_.

**Figure 8 materials-15-08281-f008:**
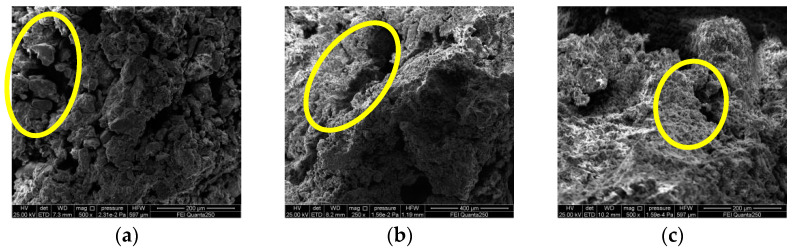
SEM image of dry shrinkage test. (**a**) A_1_B_1;_ (**b**) A_1_B_2;_ (**c**) A_1_B_3;_ (**d**) A_2_B_1;_ (**e**) A_2_B_2;_ (**f**) A_2_B_3;_ (**g**) A_3_B_1;_ (**h**) A_3_B_2;_ (**i**) A_3_B_3_.

**Figure 9 materials-15-08281-f009:**
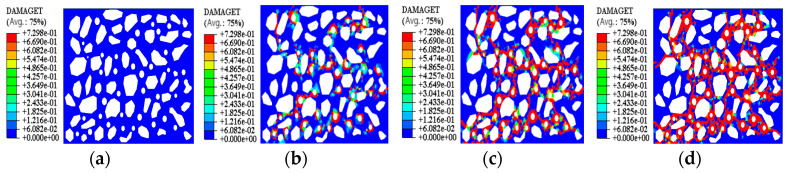
Damage nephogram of model 1 at different stages. (**a**) Step Time = 0; (**b**) Step Time = 0.37; (**c**) Step Time = 0.52; (**d**) Step Time = 1.

**Figure 10 materials-15-08281-f010:**
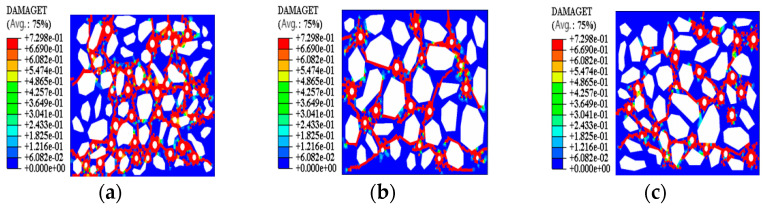
Comparison of damage nephogram of model 1–3 with test results. (**a**) Model 1; (**b**) Model 2; (**c**) Model 3.

**Figure 11 materials-15-08281-f011:**
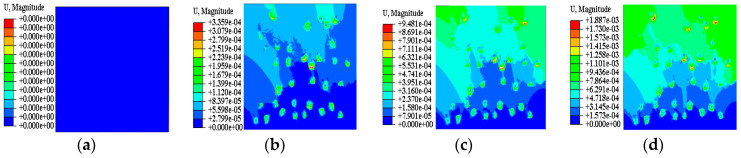
Axial displacement nephogram of model 1 at different stages. (**a**) Step Time = 0; (**b**) Step Time = 0.37; (**c**) Step Time = 0.52; (**d**) Step Time = 1.

**Figure 12 materials-15-08281-f012:**
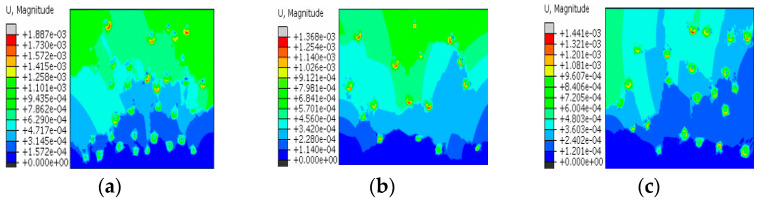
Model 1–3 axial displacement nephogram. (**a**) Model 1; (**b**) Model 2; (**c**) Model 3.

**Figure 13 materials-15-08281-f013:**
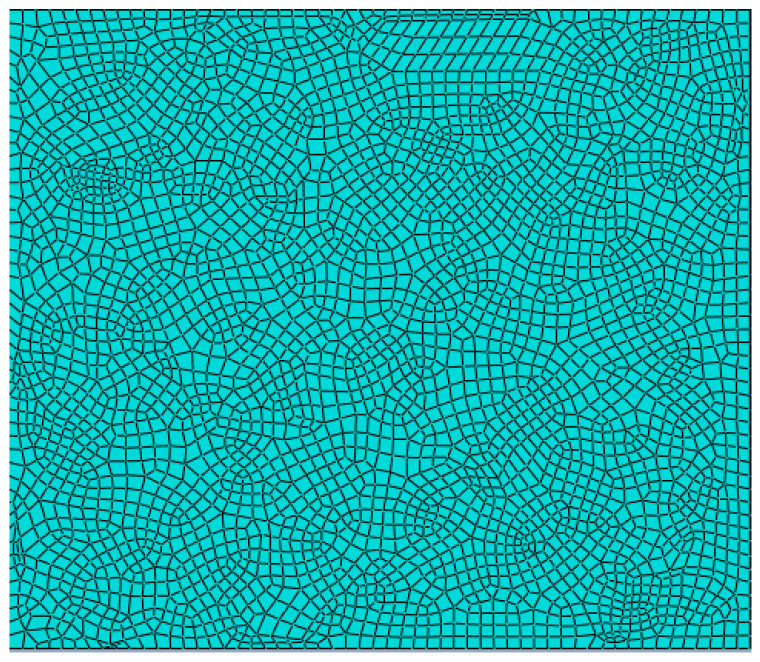
Meshing results.

**Table 1 materials-15-08281-t001:** Basic properties of soil samples from the Zhouqiao site.

Location of Soil Collection	Liquid Limit/%	Plastic Limit /%	Plasticity Index	Natural Dry Density /(g/cm^3^)	Natural Water Content/%
Upper Level	35.5	21.2	14.3	1.63	12.5
Middle Level	32.8	19.7	13.1	1.65	14.3
lower level	30.9	18.5	12.4	1.67	15.6

**Table 2 materials-15-08281-t002:** Dry shrinkage test program.

Test Number	Control Variables		Specimen Combination
	Thickness of Soil Layer (A)	Water Content (B)	
1	1 cm	20%	A_1_B_1_
2		28%	A_1_B_2_
3		35%	A_1_B_3_
4	3 cm	20%	A_2_B_1_
5		28%	A_2_B_2_
6		35%	A_2_B_3_
7	5 cm	20%	A_3_B_1_
8		28%	A_3_B_2_
9		35%	A_3_B_3_

**Table 3 materials-15-08281-t003:** Dry shrinkage test results.

Specimen Number	A_1_B_1_	A_1_B_2_	A_1_B_3_	A_2_B_1_	A_2_B_2_	A_2_B_3_	A_3_B_1_	A_3_B_2_	A_3_B_3_
Slit rate/%	9.1	7.9	7.8	6.8	4.4	6.9	3.7	2.8	1.2

**Table 4 materials-15-08281-t004:** SEM image fracture area ratio.

Group	A_1_B_1_	A_1_B_2_	A_1_B_3_	A_2_B_1_	A_2_B_2_	A_2_B_3_	A_3_B_1_	A_3_B_2_	A_3_B_3_
Fracture area ratio	0.36	0.33	0.29	0.24	0.21	0.29	0.18	0.11	0.04

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
