# Peer review of "Study on Cracking Law of Earthen Soil under Dry Shrinkage Condition"

_materials, 2022, doi:10.3390/ma15238281_

Round 1

Reviewer 1 Report

On this occasion I would like to congratulate the team of researchers for the very interesting article.

I would like the authors to make the following corrections to the article (which are intended to increase the value of the article):

1. please reinsert figure 5 in the article - it is not clear

2. the corresponding bibliographic reference for equation 1 and figure 5 is not specified

3. I request that the authors to expand the conclusions. 

Reviewer 2 Report

The paper investigates an interesting and novel topic and it is well introduced. Methodology is appropriated and English is correct. Several points need to be improved.

1) How the tests were selected? Please specify how the values in table 2 were defined

2) Equation 1 is not clear: it needs to be rewritten

3) Section 2.3. The numerical model needs to be described in details: type and number of elements, the choice of mesh, convergence procedure, boundary conditions.

4) Section 3.3 regards the numerical model results. Please make a new section n. 4. This will make the structure of the paper more balanced.

Major revisions required

Round 2

Reviewer 2 Report

the reviewer has really appreciated the answers that helped to improve the paper. However there are still two important clarifications:

1. the authors wrote: "affected by rainfall, the water content of the site varies in different depths. In particular, there is a water pit in the middle of the site, and the water content of the soil has reached the liquid limit, which is also considered in the test design". 

The role of the water level is very important, please clarify and details how it was considered, by referring to this paper:

Forcellini D. (2020) “The Role of the Water Level in the Assessment of Seismic Vulnerability for the 23 November 1980 Irpinia-Basilicata Earthquake”. Geosciences 2020;10(6):229. https://doi.org/10.3390/geosciences10060229.      2. The numerical model needs to be described in details, you may consider these references:    Forcellini D, Gobbi S (2015) Soil structure interaction assessment with advanced numerical simulations. In: Computational methods in structural dynamics and earthquake engineering (COMPDYN) conference, Crete Island, Greece, 25–27 May 2015. 
